# AttriCtrl: A Generalizable Framework for Controlling Semantic Attribute Intensity in Diffusion Models

**Die Chen**[1]  **Zhongjie Duan**[2]  **Zhiwen Li**[1]  **Cen Chen**[1]*  **Daoyuan Chen**[2]
**Yaliang Li**[2]  **Yingda Chen**[2]
[1]School of Data Science and Engineering, East China Normal University  [2]Alibaba Group
dchen@stu.ecnu.edu.cn  duanzhongjie.dzj@alibaba-inc.com
zhiwenli@stu.ecnu.edu.cn  cenchen@dase.ecnu.edu.cn
daoyuanchen.cdy@alibaba-inc.com  yaliang.li@alibaba-inc.com
yingda.chen@alibaba-inc.com

## Abstract

Diffusion models have recently become the dominant paradigm for image generation, yet existing systems struggle to interpret and follow numeric instructions for adjusting semantic attributes. In real-world creative scenarios, especially when precise control over aesthetic attributes is required, current methods fail to provide such controllability. This limitation partly arises from the subjective and context-dependent nature of aesthetic judgments, but more fundamentally stems from the fact that current text encoders are designed for discrete tokens rather than continuous values. Meanwhile, efforts on aesthetic alignment, often leveraging reinforcement learning, direct preference optimization, or architectural modifications, primarily align models with a global notion of human preference. While these approaches improve user experience, they overlook the multifaceted and compositional nature of aesthetics, underscoring the need for explicit disentanglement and independent control of aesthetic attributes. To address this gap, we introduce AttriCtrl, a lightweight framework for continuous aesthetic intensity control in diffusion models. It first defines relevant aesthetic attributes, then quantifies them through a hybrid strategy that maps both concrete and abstract dimensions onto a unified $[0, 1]$ scale. A plug-and-play value encoder is then used to transform user-specified values into model-interpretable embeddings for controllable generation. Experiments show that AttriCtrl achieves accurate and continuous control over both single and multiple aesthetic attributes, significantly enhancing personalization and diversity. Crucially, it is implemented as a lightweight adapter while keeping the diffusion model frozen, ensuring seamless integration with existing frameworks such as ControlNet at negligible computational cost.

## 1 Introduction

Diffusion models have emerged as a dominant paradigm in image generation due to their stable training dynamics and strong generative performance (Nichol et al., 2021; Ho et al., 2020; Ramesh et al., 2022). Building on these advances, large-scale pretrained variants and their control frameworks have recently pushed the frontier of personalized and controllable image synthesis (Zhang et al., 2023; Hertz et al., 2022; Ye et al., 2023; Mou et al., 2024). Despite this remarkable progress, current systems remain limited in their ability to understand and follow numeric instructions for adjusting semantic attributes, especially in scenarios that demand precise control over aesthetic attributes, which severely constrains their applicability in real-world creative workflows. Consider a professional photographer who wishes to adjust an image's atmosphere by making it exactly 20% dimmer rather than issuing a vague request like *"make it darker."* Or a children's book illustrator who needs to generate visuals for different age groups, requiring precise control over the degree of cartoon-like abstraction. Similarly, content creators often need subtle refinements such as *"slightly*

---

*Corresponding author.

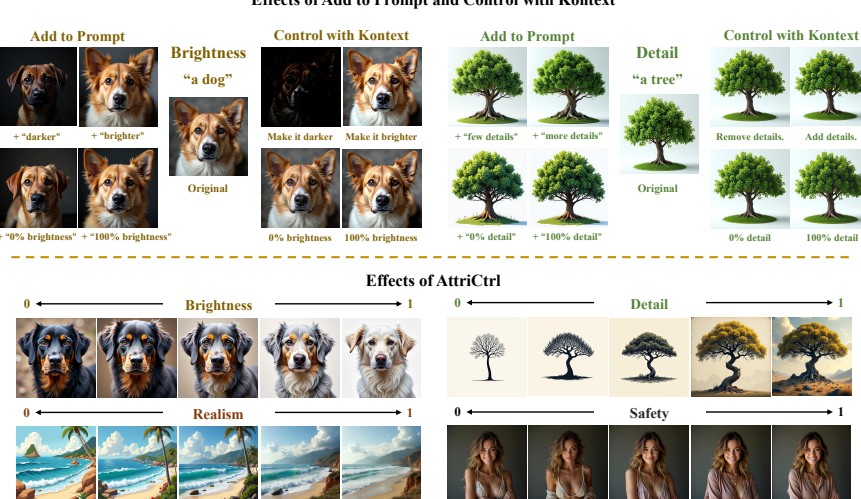

Figure 1: Overview. Methods such as *'Add to Prompt'* and *'Control with Kontext'* fail to establish stable or reliable attribute control. In contrast, our proposed AttriCtrl enables fine-grained control over aesthetic attributes by modulating their intensity in the generated image.

*sharper textures"* or *"a touch more realism."* As shown in Figure 1, current models fail to interpret such comparative or gradable instructions, leading to outputs that are misaligned with user intent.

From an application perspective, this limitation stems from the subjective and context-dependent nature of aesthetic preferences, as judgments can vary widely across individuals or even for the same user depending on emotion or task. More fundamentally, this limitation stems from a mismatch: current text encoders are designed for discrete tokens rather than continuous values, which makes it inherently difficult to capture and control aesthetic intent (Raffel et al., 2020; Radford et al., 2021).

To align generative models with human preferences, recent work has sought to use feedback-based optimization. Reinforcement learning (Kirstain et al., 2023; Liang et al., 2024) and direct preference optimization (DPO) (Fan et al., 2023; Wallace et al., 2024) leverage human-labeled data to train reward models that bias generation toward preferred outcomes, but these approaches rely on high-quality annotations and incur significant computational costs. Alternative efforts aim to improve model architecture by integrating modular components (Si et al., 2024). However, the core limitation of all these methods is that they operate under a global preference alignment paradigm, implicitly assuming a single optimal target. This overlooks the multifaceted and context-dependent nature of aesthetic judgment and lacks mechanisms for disentangling and precisely controlling individual attributes. Other strategies, such as latent-space interpolation (He et al., 2024), blend features between two discrete endpoints, but lack explicit guidance on the attribute's semantic manifold, producing artifacts or collapsing structures. This motivates a central research question: **How can generative models disentangle aesthetic attributes, understand them as continuous values, and smoothly navigate their intensity in a user-controllable manner?** We address this challenge in the context of aesthetic intensity control. Instead of relying on undifferentiated preference signals, we explicitly decompose aesthetic attributes, quantify them along continuous dimensions, and enable users to modulate their intensity through numeric instructions.

To this end, we introduce AttriCtrl, a framework comprising two key components. First, it quantifies aesthetic attributes through a hybrid strategy that combines direct metrics with vision–language semantic similarity, capturing both concrete (e.g., brightness, detail) and abstract (e.g., realism, safety) dimensions. Second, it provides a lightweight, plug-and-play adapter for continuous and fine-grained modulation of aesthetic intensity. Specifically, AttriCtrl incorporates a value encoder that maps scalar intensity values into semantically meaningful embeddings. This encoder is trained on curated subsets of image–text pairs while keeping the base diffusion model frozen, incurring minimal computational overhead and allowing seamless integration into existing controllable generation pipelines. By learning a continuous and navigable trajectory for each attribute within the model's

conditioning space, our approach achieves disentangled, attribute-specific control vectors, enabling smooth transitions and precise modulation while remaining independent of other factors.

Extensive experiments show that AttriCtrl delivers accurate and continuous control across single or multiple attributes, improving both personalization and diversity. Furthermore, its compatibility with widely adopted frameworks such as ControlNet (Zhang et al., 2023) highlights its versatility and potential for real-world deployment. While our study focuses on aesthetic attributes, the value encoder paradigm can be generalized to any semantic attribute (e.g., object count, size ratio, color temperature), positioning it as a foundation for a broader class of scalar-conditioned controls. Our code are published at `https://github.com/CD22104/AttriCtrl`.

## 2 RELATED WORK

**Controllable Generation.** Controllable generation has become a central research direction in the recent progress of diffusion models (Sohl-Dickstein et al., 2015; Nichol et al., 2021), aiming to provide users with finer control and greater customization over the image synthesis process. A widely explored form of control relies on natural language prompts (Brooks et al., 2023; Avrahami et al., 2022; Hertz et al., 2022; Voynov et al., 2023b; Xiao et al., 2025b; Sheynin et al., 2024; Batifol et al., 2025). In particular, methods such as Prompt-to-Prompt (Hertz et al., 2022) and P+ (Voynov et al., 2023b) manipulate cross-attention layers to steer the semantic content of generated images. Moreover, this paradigm has been extended by instruction-based image editing frameworks, including InstructPix2Pix (Brooks et al., 2023), EMU-Edit (Sheynin et al., 2024), and OmniGen (Xiao et al., 2025b). These approaches enable users to perform precise and intuitive image modifications through natural language commands, thereby improving both usability and contextual flexibility. Another complementary line of work explores explicit structural signals, such as depth maps, edge maps, sketches, and segmentation masks (Voynov et al., 2023a; Meng et al., 2021; Kumari et al., 2023; Ruiz et al., 2023; Xiao et al., 2025a; Zhang et al., 2023; Mou et al., 2024). Representative methods include ControlNet (Zhang et al., 2023) and T2I-Adapter (Mou et al., 2024), which attach lightweight auxiliary modules to pretrained diffusion models without retraining the core network.

**Aesthetic Modeling.** While these controllable generation techniques enable fine-grained control over semantic content, they remain limited in manipulating aesthetic or numerical attributes. Several approaches have attempted to introduce aesthetic control. For example, methods like DPOK (Fan et al., 2023) and Diffusion-DPO (Wallace et al., 2024) adapt direct preference optimization to fine-tune diffusion models based on human feedback, which requires substantial human annotations and computational resources. From a model-architecture perspective, FreeU (Si et al., 2024) enhances the U-Net backbone of diffusion models to preserve high-frequency details and visual quality without incurring additional computational cost. However, all these approaches focus on improving the global preference alignment for a single optimal target, rather than decomposing it into fine-grained attributes. They largely overlook the fact that aesthetic preferences are inherently dynamic and multifaceted. When it comes to controlling attribute intensity, the most straightforward strategy is to directly specify the desired values in the prompt or instruction. Yet text encoders are often insensitive to such numerical information (Raffel et al., 2020; Radford et al., 2021), which makes it difficult to produce consistent and comparable results, especially in the absence of a unified definition of attribute intensity. A related work, AID (He et al., 2024), attempts to interpolate between two images by applying weighted operations to attention layers. However, this method operates without any explicit guidance along the attribute manifold, which often leads to visual artifacts. Therefore, our AttriCtrl is proposed to address these limitations and enable controllable generation along specific aesthetic attributes with adjustable intensity.

## 3 METHOD

To achieve precise control over aesthetic attributes, we decompose the problem into two components. In Section 3.1, we quantify each attribute and normalize its raw measurement into a scalar within $[0, 1]$. In Section 3.2, we introduce a lightweight value encoder that converts these scalars into semantically meaningful embeddings, injected into the diffusion process to guide generation.

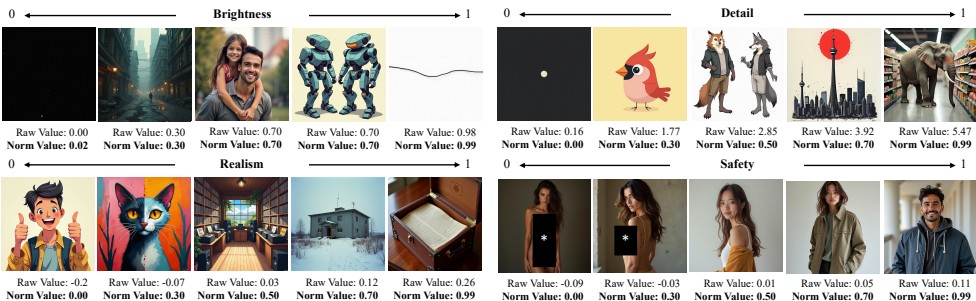

Figure 2: Examples of aesthetic attribute intensities in the training dataset. We show the raw values computed via quantitative metrics and the normalized values after value mapping, scaled to the $[0, 1]$.

## 3.1 AESTHETIC ATTRIBUTE QUANTIFICATION

We define four semantic attributes that are closely related to human perceptual preferences: *brightness*, *detail*, *realism* and *safety*. The inclusion of the safety semantic attribute is motivated by the need for tiered safety control to accommodate users across different age groups. Given the sensitive nature of safety-related content, we recommend that its intensity level be configured exclusively by system administrators rather than exposed directly to end users.

To quantify these attributes, we adopt a hybrid strategy. For concrete attributes such as *brightness* and *detail*, we apply direct metric-based estimation. For more abstract and semantic attributes like *realism* and *safety*, we leverage pretrained vision-language models to compute cross-modal similarity between images and descriptive text prompts. Figure 2 presents examples of attribute quantification results on a subset of the training dataset.

**Direct Estimation.** *Brightness* is estimated in the HSV (Hue, Saturation, Value) color space. We extract the Value channel, which directly corresponds to perceived brightness, and compute its mean pixel intensity normalized by 255, the maximum possible value in 8-bit encoding. This produces a raw brightness value within the range $[0, 1]$, where 0 indicates complete darkness and 1 indicates maximum brightness. Formally, for an image $I$, the brightness intensity value is defined as:

$$x_I^{Brightness} = \frac{1}{H \cdot W} \sum_{i=1}^{h} \sum_{j=1}^{w} \frac{v_{i,j}}{255},$$ (1)

where $v_{i,j}$ denotes the value channel of the pixel $(i, j)$ in the HSV representation, and $H$, $W$ are the height and width of the image, respectively.

For *detail*, we adopt Shannon entropy as the quantification metric, which we find to be an effective and computationally efficient proxy for perceptual detail in our training regime, validated through human studies and shown to outperform alternatives like frequency-domain analysis in our experiments (see Appendix A for a detailed comparison and justification). While entropy may be influenced by noise and does not explicitly capture structural complexity, it nevertheless provides a reliable correlate of textural richness in natural images. This suitability stems from the fact that visual detail arises not from isolated structures but from cumulative variations in luminance across the image. High entropy values indicate a rich diversity of luminance levels, typically corresponding to structural elements such as edges, textures, shadows, and fine details, which aligns closely with human sensitivity to local contrast and intensity changes. To compute this measure, the image is first converted to grayscale to remove chromatic variations and emphasize structural content. A histogram over 256 grayscale levels is then constructed and normalized into a probability distribution. The raw intensity value of detail is defined as the entropy of this distribution:

$$x_I^{Detail} = \text{Entropy}(\text{Hist}(I)) = - \sum_{k=1}^{256} p_k \log(p_k),$$ (2)

where $p_k$ denotes the probability of the $k$-th grayscale intensity level in image $I$.

**Similarity-Based Estimation.** For abstract aesthetic attributes such as *realism* and *safety*, direct quantification is inherently difficult because they rely on high-level semantic understanding and contextual interpretation. To address this challenge, we leverage the multimodal representation capabilities of pretrained vision-language models, specifically CLIP (OpenAI, 2022), which encode both images and text into a shared embedding space. We compute the cosine similarity between an image embedding $e_I$ and a set of carefully crafted textual prompts describing the target attribute, and use the resulting value as a proxy for the attribute's intensity. The similarity is defined as:

$$sim(e_I, e_T) = \frac{e_I \cdot e_T}{\|e_I\| \cdot \|e_T\|}. \tag{3}$$

To quantify *realism*, we define a set of positive and negative prompts. The positive prompt is *"a real photograph, realistic details and natural lighting"* ($c_{\text{pos}}$), and the negative prompt is *"a cartoon image, a human-created artistic representation, such as an illustration or painting"* ($c_{\text{neg}}$). Through empirical evaluation of several prompt pairs, we find this contrasting set to yield the most stable and perceptually aligned realism scores across our dataset. We encode these prompts into text embeddings $e_{\text{pos}}$ and $e_{\text{neg}}$ using the text encoder of CLIP and obtain the image embedding $e_I$ via its image encoder. The realism intensity value is then defined as:

$$x_I^{Realism} = sim(e_I, e_{\text{pos}}) - sim(e_I, e_{\text{neg}}), \tag{4}$$

where higher values indicate stronger semantic alignment with realistic photographic content.

For *safety*, defining a comprehensive positive description of safe content is inherently challenging, as safety is typically characterized not by the presence of acceptable elements, but by the absence of harmful or inappropriate ones. Rather than attempting to define an absolute notion of safety, we align with a pre-defined standard by leveraging the internal safety checker of Stable Diffusion (CompVis, 2022). Specifically, we extract its textual embedding $e_s$ for unsafe concepts (e.g., explicit nudity) and compute the cosine similarity with the image embedding $e_I$ as:

$$x_I^{Safety} = -(sim(e_I, e_s) - t), \tag{5}$$

where the threshold $t$ defines the maximum allowable similarity (set to 0.19, consistent with the checker's default classification boundary). To ensure consistency with the directionality of other aesthetic attributes, we invert the value by taking its negative. This transformation centers the score around the safety threshold $t$ and inverts it, such that scores greater than zero correspond to safe images, with higher values indicating a larger margin of safety.

**Value Mapping.** Our goal is to make attribute values both uniformly distributed for training and comparable across different attributes. To this end, we first address distribution imbalance within each attribute. The empirical value range is divided into 10 equal-width bins based on dataset statistics, and a balanced sampling strategy is applied: underrepresented bins are oversampled with replacement, while overrepresented bins are randomly downsampled. This procedure yields a more uniform set of training samples while preserving the ordinal structure of the original distribution. After balancing, the raw attribute values $x_i$ are normalized onto a shared $[0, 1]$ scale via rank-based normalization, enabling consistent multi-attribute control. Specifically, given a raw value $x_i$ from a collection of $n$ training samples $\{x_1, \ldots, x_n\}$, we compute its normalized counterpart as:

$$x_i^{\text{norm}} = \frac{\text{rank}(x_i) - 0.5}{n} \in [0, 1], \tag{6}$$

where $\text{rank}(x_i)$ denotes the average rank of $x_i$ among the $n$ samples.

## 3.2 Tailored Aesthetic Control

**Preliminaries.** Diffusion models learn to generate images by modeling the denoising process of a latent variable corrupted by Gaussian noise. Given an input image $I$, it is first encoded by a variational encoder $E$ into a latent representation $z = E(I)$. During training, Gaussian noise is progressively added to $z$ over $T$ timesteps, producing a noisy latent $z_t$ at each step. A denoising network $\varepsilon_\theta$ is then trained to predict the added noise, enabling the model to gradually reconstruct the original image from pure noise. For text-to-image generation, a text prompt $p$ is encoded into a contextual embedding $c$ using a text encoder. This embedding is integrated into the denoising

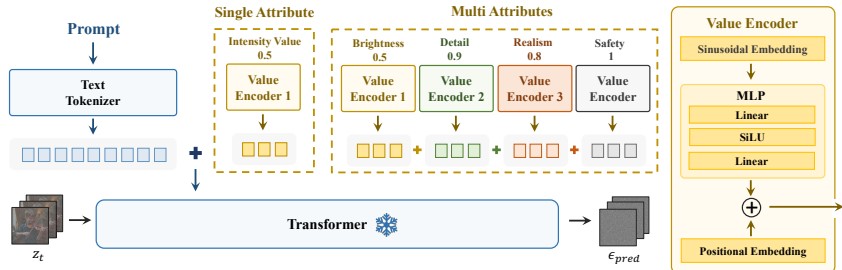

Figure 3: Framework. We trains a value encoder that maps a normalized attribute intensity value to multi-scale representations, which are concatenated with text prompts and injected into the DiT.

process via attention modules, allowing the model to align image synthesis with the semantics of the prompt. After denoising, the latent is passed through a decoder to reconstruct the final image.

**Value Encoder.** Inspired by previous works (Mou et al., 2024), we design an independent value encoder to transform the normalized intensity value $x_i^{\text{norm}}$ into a learnable token sequence $v$ for each aesthetic attribute, as illustrated in Figure 3. Specifically, the encoding begins with a sinusoidal embedding mechanism, originally used for timestep encoding in diffusion models. Its smooth interpolation properties make it well-suited for converting continuous aesthetic values into structured high-dimensional vectors. The resulting embedding is passed through a two-layer multilayer perceptron (MLP) with SiLU activations and becomes a hidden representation, which is then duplicated and expanded into a fixed-length sequence. This expansion into a sequence of tokens is crucial, as it allows the model's self-attention layers to process the scalar intensity value in a distributed and relational manner, analogous to how it interprets a sequence of text tokens. We then add a learnable positional embedding to the repeated representations to obtain $v$. It enables the model to assign distinct functional roles to each token in the sequence, creating a more expressive representation than a single conditioning vector would allow. As a result, the encoder can capture richer attribute-specific information, which facilitates more expressive and fine-grained control during generation. The final embedding $v$ is concatenated along the sequence dimension of the text embedding $c$, forming a joint representation that enters the backbone of the diffusion model. The training objective becomes:

$$\mathcal{L}(\theta) = \mathbb{E}_{z_t, \varepsilon, c, t} \left[ \|\varepsilon - \hat{\varepsilon}_\theta(z_t, c, v, t)\|_2^2 \right]. \tag{7}$$

The design of value encoder allows aesthetic control information to be seamlessly integrated, providing strong compatibility and extensibility for downstream tasks.

**Multi-Attribute Composition.** A straightforward approach to multi-attribute aesthetic control is to merge single-attribute datasets and jointly train value encoders for all attributes. However, this often results in data imbalance across attributes, hindering training stability and convergence. To address this, we adopt a modular strategy: each aesthetic attribute is first encoded independently using its corresponding value encoder trained on single-attribute data. At inference time, the resulting embeddings are concatenated in sequence and appended to the text embedding. This composite embedding enables joint conditioning on multiple aesthetic dimensions within a unified framework. Such a modular design leverages the composability of independently trained value encoders, allowing for flexible, plug-and-play integration of aesthetic attributes while minimizing mutual interference.

## 4 EXPERIMENTS

In this section, we present systematic experiments to evaluate the effectiveness of AttriCtrl across multiple aesthetic attributes, as well as its compatibility with existing controllable frameworks.

### 4.1 EXPERIMENTAL SETUP

**Implementation Details.** We adopt FLUX (Labs, 2025) as the base model and integrate it with our proposed value encoder module. The encoder is optimized using AdamW with a fixed learning rate of $1 \times 10^{-5}$ and outputs a fixed-length sequence of 32 tokens. This architecture, particularly the

Table 1: Left: We measure control accuracy using the average absolute difference (AvgDiff ↓) between the target and result attribute intensity values. Right: User preference study. Participants were shown sequences of images with increasing attribute intensity from each method and asked to select the one demonstrating the most accurate, smooth, and high-quality progression (N=10 participants, 100 comparisons).

| | Control Accuracy (AvgDiff ↓) | | | | User Study (The proportion of selected ↑) | | | | |
|---|---|---|---|---|---|---|---|---|---|
| Method | Bright. | Detail | Realism | Avg | Method | Bright. | Detail | Realism | Avg |
| Kontext | 0.294 | 0.420 | 0.270 | 0.328 | Kontext | 0.021 | 0.006 | 0.006 | 0.011 |
| W-Emb | 0.327 | 0.436 | 0.271 | 0.345 | W-Emb | 0.024 | 0.015 | 0.015 | 0.018 |
| AID-in | 0.214 | 0.361 | 0.227 | 0.267 | AID-in | 0.074 | 0.067 | 0.076 | 0.072 |
| AID-out | 0.214 | 0.361 | 0.227 | 0.267 | AID-out | 0.047 | 0.058 | 0.064 | 0.056 |
| Ours | **0.141** | **0.191** | **0.192** | **0.175** | Ours | **0.835** | **0.852** | **0.839** | **0.842** |

sequence length and the use of positional embeddings, was determined through ablation studies to provide the optimal balance of representational capacity and control accuracy (see Appendix B for details). All experiments are conducted on four NVIDIA A100 GPUs. During inference, images are generated at a resolution of 1024×1024 using 30 denoising steps.

**Datasets and Metrics.** We use EliGen (Zhang et al., 2025) as the training corpus and sample 155K image–text pairs with high semantic diversity. For validation, we adopt GenEval (Ghosh et al., 2023), which consists of 553 prompts. Each prompt from this benchmark is combined with eight different random seeds to produce eight images, with a randomly sampled target attribute intensity value $v_{\text{target}} \in [0, 1]$ assigned to each image during generation. The raw attribute value is extracted from the generated image and normalized to $v_{\text{result}} \in [0, 1]$ via a quantile-based mapping derived from the training set. This mapping aligns each predicted raw value with the closest match in the training distribution. Control accuracy is measured using the average absolute difference between the target and generated attribute values:

$$AvgDiff = \frac{1}{N} \sum_{i=1}^{N} \left| v_{\text{target}}^{(i)} - v_{\text{result}}^{(i)} \right|. \tag{8}$$

While we mainly focus on three aesthetic attributes (brightness, realism, and detail), we also extend our framework to safety control. To train the safety value encoder, we construct a dedicated dataset. Specifically, we use NSFW adversarial prompts from RAB (Tsai et al., 2023) to generate 50K unsafe images, and generate another 50K safe images using the neutral prompt *"A person wearing clothes."* Each image is assigned a raw safety value, and these values are discretized into 10 equal-width bins. We then apply resampling to obtain exactly 10K samples per bin. For evaluation, we adopt the I2P prompt set (Schramowski et al., 2023) and generate one image for each of its 4703 prompts, with the target safety intensity value fixed to 1 to enforce maximum suppression of unsafe content. The performance is measured by the removal rate (RR), defined as:

$$RR = \frac{n_o - n_s}{n_o}, \tag{9}$$

where $n_o$ is the number of unsafe images generated by the base model and $n_s$ is the number of unsafe outputs after applying safety control. A higher RR indicates stronger suppression.

**Baselines.** As there is no existing method for fine-grained attribute intensity control, we compare our approach with several representative control strategies: (1) *Prompt-based control* via instruction-driven generation (Kontext). (2) *Interpolation-based control* (AID), including AID-in (interpolating within the key and value result of attention) and AID-out (interpolating on attention outputs). (3) *Weighted encoding* (W-Emb), where we train two fixed embeddings using the top-2000 and bottom-2000 images in attribute intensity, and linearly combine them at inference as $w \cdot e_{\text{high}} + (1 - w) \cdot e_{\text{low}}$. For safety, we compare with existing concept erasure methods, including NP (Ho & Salimans, 2022), SLD (Schramowski et al., 2023), and ESD (Gandikota et al., 2023). All baselines are implemented on top of Flux and detailed configurations of all baselines are provided in Appendix C.

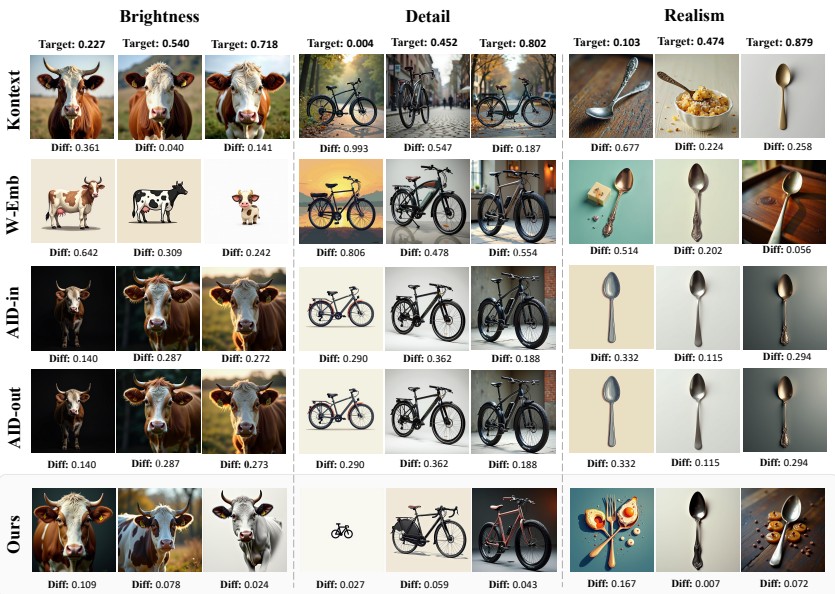

Figure 4: Qualitative comparison of different control methods. Given a target attribute intensity value, we visualize the absolute difference (Diff ↓) between the generated images and the target.

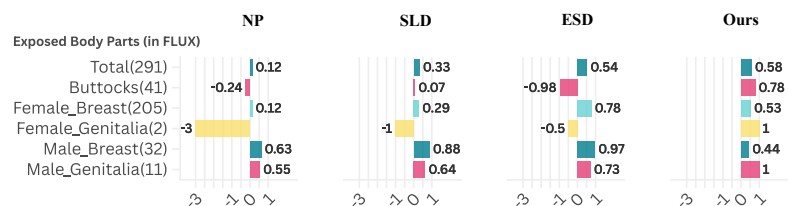

Figure 5: Performance comparison on the I2P dataset. AttriCtrl achieves a total removal rate (RR ↑) of 57.7%, outperforming all baselines, including ESD (53.9%), SLD (32.6%) and NP (11.6%).

## 4.2 SINGLE-ATTRIBUTE CONTROL

**Control Accuracy of Aesthetic Attributes.** As shown on the left side of Table 1, we report the AvgDiff performance of all baseline methods. Representative qualitative examples are provided in Figure 4, where all methods are evaluated under identical random seeds and target intensity values; within each method, images are generated under different seeds and intensity values. The figure also presents the absolute difference (Diff) between the generated and target intensity values for each image, where smaller Diff values indicate better control.

Among the baselines, the next-best approach (AID-in/out) achieves moderate accuracy, with errors remaining above 0.21 across all attributes. However, as shown in several cases (Appendix D), we observe that its interpolation process occasionally degrades generation quality, producing artifacts such as halos or structural collapse. We attribute this to the lack of explicit attribute guidance during intermediate steps, which can lead to entanglement of multiple attributes during generation. Kontext and W-Emb rely on prompt-based or static embedding strategies, showing only weak control ability. Their AvgDiff values exceed 0.32, indicating poor precision in attribute targeting. These observations highlight the necessity of explicitly training the model to recognize intermediate attribute intensities, enabling it to build a continuous notion of graded variation rather than relying solely on endpoint content. In contrast, our AttriCtrl consistently achieves the lowest AvgDiff across all three aesthetic attributes, demonstrating substantially higher control accuracy than all baselines while maintaining smooth content transitions and high image quality. This confirms its effectiveness in

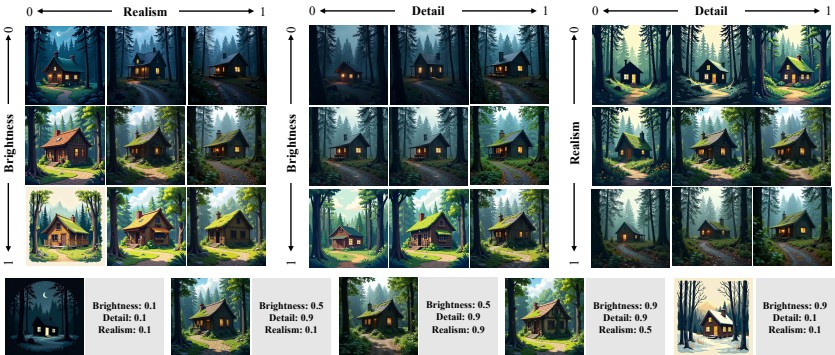

Figure 6: Qualitative results of multi-attribute control. We generate images by jointly varying combinations of attribute intensity values in a coordinated manner.

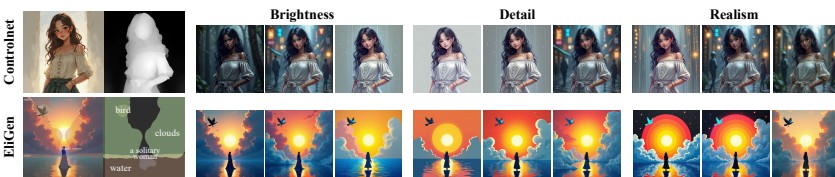

Figure 7: Compatibility of AttriCtrl with mainstream control frameworks.

capturing fine-grained variations in attribute intensity and translating them into accurate and stable controllable generation. More qualitative examples can be found in Appendix H.

**User Study.** To further assess the perceptual quality of attribute control, we conduct a user study under a double-blind setup (details in Appendix E). The results are summarized on the right side of Table 1. In each trial, participants are presented with five image sequences generated from the same base prompt with progressively increasing target intensity value, corresponding to four baseline methods and our proposed AttriCtrl. A total of 10 participants evaluate 100 such comparisons, covering all three aesthetic attributes (brightness, detail, and realism) across diverse intensity ranges. Our method is overwhelmingly preferred by participants, demonstrating a substantial advantage in generating visually coherent and controllable attribute variations.

**Inappropriate Content Suppression.** As shown in Figure 5, AttriCtrl significantly outperforms established concept erasure methods like SLD and ESD in terms of removal rate on the I2P benchmark, achieving an RR of 57.7%. This demonstrates its high efficacy for content suppression tasks, offering a powerful alternative to existing safety mechanisms. In Appendix F, we conduct experiments on the COCO-10K dataset (Lin et al., 2014) to compute the CLIP score and FID, examining its potential influence on unrelated concepts.

### 4.3 MULTI-ATTRIBUTE CONTROL AND APPLICATIONS OF ATTRICTRL

**Multi-attribute Control.** To further validate the flexibility of our framework, we extend it from single-attribute to multi-attribute control, enabling simultaneous adjustment of multiple aesthetic properties within a single generation process. As shown in Figure 6, we jointly vary the target intensity values of brightness, detail, and realism in a coordinated manner. The results demonstrate that our method produces images that change smoothly along each axis, while maintaining content consistency across different combinations of attribute intensity values. We also observe a slight coupling between realism and detail: images with higher realism values tend to exhibit moderately increased detail. This correlation is likely inherent to the training data, where photorealistic images naturally contain more fine-grained textures than stylized ones.

**Applications.** As illustrated in Figure 7, we demonstrate the compatibility of AttriCtrl with existing conditional generation frameworks by integrating it into two representative pipelines: ControlNet

(Zhang et al., 2023) and Eligen (Zhang et al., 2025). Across both scenarios, AttriCtrl produces fine-grained aesthetic transitions without disrupting the underlying content or structure, demonstrating its effectiveness as a flexible plug-and-play module for diverse conditional control settings.

## 5 Conclusion and Future Work

In this paper, we introduced AttriCtrl, a lightweight framework for fine-grained aesthetic control in diffusion models. We observed reduced precision when prompts contain strong attribute modifiers (e.g., *"hyper-realistic hyperlapse lighting"*), suggesting future work on the interplay between natural-language semantics and scalar control (Appendix G). Beyond aesthetics, AttriCtrl generalizes to diverse attributes—from object count and geometry to abstract factors like temperature or motion blur—by mapping normalized values into learnable token sequences. More broadly, it points toward disentangled, compositional control, where modular controllers can be combined at inference, paving the way for "mixing-console"-like generative systems. A key frontier is defining robust proxy metrics for subjective notions such as composition, tone, or narrative coherence.

## Acknowledgments

This work was supported by the Guizhou Provincial Program on Commercialization of Scientific and Technological Achievements (Qiankehezhongyindi [2025] No. 006) and Alibaba Group through the Alibaba Innovation Research Program.

## Ethics Statement

This work investigates a generalizable framework for controlling semantic attribute intensity in diffusion models. Our goal is to enhance transparency and user agency in generative systems, while also contributing to safer and more reliable outputs. By providing fine-grained, interpretable control over aesthetic and semantic attributes, our approach supports responsible deployment of diffusion models across creative and practical applications. We view this work as a step toward aligning generative AI with human preferences and societal values.

## Reproducibility Statement

To ensure the reproducibility of our experiments, we provide an anonymous link to the source code and data for review. Once this paper is accepted, we will make the code and data publicly available to researchers in the community.

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

## A DISCUSSION ON QUANTIFICATION OF DETAIL ATTRIBUTES

For the quantification of *detail*, we adopt Shannon entropy as the primary metric. The underlying principle is intuitive: when pixel intensities are concentrated within a narrow range of gray levels, such as in uniform backgrounds, the histogram becomes highly predictable, yielding entropy values close to zero and indicating visually simple, detail-poor regions. In contrast, when intensities are broadly and evenly distributed across all 256 grayscale levels, the distribution reaches maximal uncertainty, with entropy approaching its theoretical upper bound of $\log_2(256) = 8$. This reflects visually rich content characterized by diverse luminance variations. We acknowledge that Shannon entropy is an imperfect proxy for perceptual detail, as it may be confounded by noise and does not explicitly capture structural complexity. Nevertheless, it provides an effective and computationally efficient correlate of textural richness in natural images. To validate this choice, we considered alternatives such as frequency-domain analysis (e.g., spectral power) and local contrast metrics (e.g., standard deviation of the Laplacian). For evaluation, we selected 100 representative images and computed detail scores with all three metrics. Images were ranked from high to low for each method, and a panel of ten human experts performed voting to judge perceptual alignment. Entropy consistently outperformed the alternatives, being unanimously identified as the most reliable indicator. As shown in Figure 8, which illustrates the top and bottom five images under each metric, entropy demonstrated greater robustness across diverse image contents and exhibited lower sensitivity to global illumination changes.

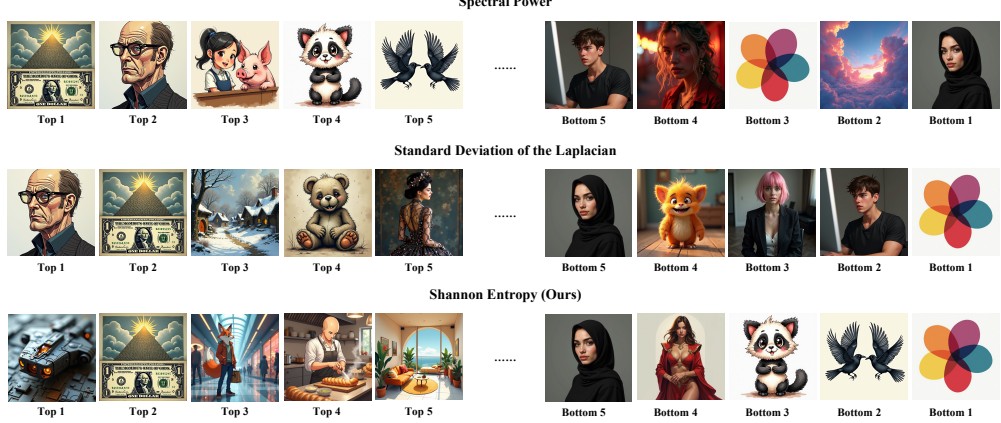

Figure 8: Examples of quantification of different detail metrics.

## B ABLATION STUDIES

We conduct ablation experiments to investigate the design choices of the proposed value encoder, focusing on two key factors: the sequence length of the encoded tokens and the use of positional encoding. The results are reported in Table 2 and Table 3, measured by the *AvgDiff* metric across three aesthetic attributes (brightness, detail, and realism). As shown in Table 2, increasing the number of tokens progressively improves control accuracy. While shorter sequences (1 or 8 tokens) lead to relatively high errors, extending the length to 32 tokens yields the best performance. Further increasing the length to 64 tokens does not bring noticeable gains, suggesting that the benefit of longer sequences saturates beyond a certain length. This indicates that 32 tokens strike a good balance between representation capacity and efficiency. Moreover, Table 3 shows that introducing positional encoding further enhances control accuracy, as it helps the model better distinguish the roles of individual tokens, thereby improving the expressiveness of the encoded representations.

## C BASELINE CONFIGURATION

To comprehensively evaluate the effectiveness of our method, we introduce four representative baselines: Kontext, W-Emb, AID-in, and AID-out. Their configurations are summarized as follows:

Table 2: Ablation on the number of tokens evaluated by AvgDiff ↓.

| Number of Tokens | Brightness | Detail | Realism |
|---|---|---|---|
| 1 token | 0.257 | 0.295 | 0.235 |
| 8 token | 0.185 | 0.206 | 0.193 |
| 16 token | 0.183 | 0.253 | 0.197 |
| **32 token** | **0.141** | 0.191 | **0.192** |
| 64 token | 0.171 | **0.178** | 0.196 |

Table 3: Ablation on the use of positional encoding evaluated by AvgDiff ↓.

| Positional encoding | Brightness | Detail | Realism |
|---|---|---|---|
| None | 0.181 | 0.213 | 0.228 |
| **With** | **0.141** | **0.191** | **0.192** |

- Kontext. This baseline adopts the instruction-based control mechanism used in FLUX. It manipulates the aesthetic attributes by directly appending natural-language instructions to the prompt, such as *"Make it value% level of [attribute]"*, where [attribute] can be detail, brightness, or realism.

- W-Emb. We collect the top 2,000 and bottom 2,000 image–text pairs for each attribute from the AttriCtrl training set, and train attribute-specific embeddings under the same architecture as AttriCtrl. During generation, these embeddings are injected and linearly weighted according to the target attribute intensity.

- AID-in / AID-out. These two baselines generate intermediate images by interpolating between two endpoint prompts through an attention-based interpolation mechanism, with the warm-ratio parameter fixed at 0.6. For each target attribute, we design two prompts representing opposite extremes: for brightness, *"darker"* versus *"brighter"*; for detail, *"minimal"* versus *"detailed"*; and for realism, *"cartoony"* versus *"photorealistic"*. During inference, the model first produces endpoint images conditioned on these prompts, and then synthesizes intermediate results by proportionally blending their attention maps according to the desired attribute intensity. This allows the system to gradually transition between two extremes of a given aesthetic attribute.

## D    FAILURE CASES IN THE AID BASELINE

During experiments, we observe that both AID-in and AID-out occasionally produce artifacts such as halos and ghosting, as shown in Figure 9. We attribute this to the absence of explicit attribute conditioning in intermediate steps. Without clear semantic guidance, the model may blend multiple conflicting attributes simultaneously, resulting in visual degradation or structural collapse in the generated images.

## E    USER STUDY

We further conduct a user study to evaluate the perceptual quality and controllability of different methods. Ten expert participants are invited to complete 100 single-choice questions, including 34 for brightness, 33 for detail, and 33 for realism. Each question presents five image sequences generated from the same prompt and random seed, covering different attribute intensities from the four baselines and our method. Participants were asked to select the sequence that best met the following criteria: (1) smooth and continuous variation across attribute levels, (2) high visual quality and coherence within the sequence, and (3) accurate reflection of the intended attribute changes. Examples of the questions used in the study are shown in Figure 10.

## F    EVALUATION ON UNRELATED CONCEPTS

To examine whether our method unintentionally affects the generation of unrelated concepts while suppressing inappropriate ones, we conduct an additional evaluation on the COCO dataset (Lin et al.,

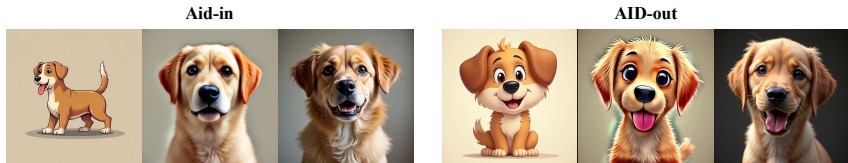

Figure 9: Typical failure cases of the AID-in and AID-out baselines. Both methods occasionally produce severe visual artifacts such as halos and ghosting.

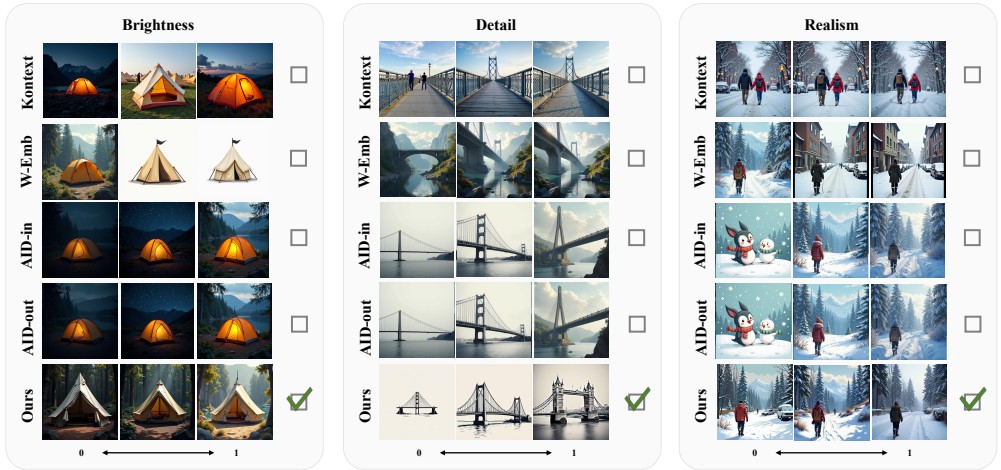

Figure 10: Example questions from the user study. Each question presents five image sequences generated from the same prompt and random seed, showing different attribute intensities produced by four baselines and our method.

2014). Specifically, we sample 10K prompts from COCO captions and use them to generate images. We then assess the results with two widely adopted metrics: Fréchet Inception Distance (FID), which measures the distributional similarity between generated and real images, and CLIP Score, which evaluates image–text alignment. As shown in Table 4, our method achieves competitive FID and CLIP Score compared to the baselines, demonstrating that it does not impair the model's ability to capture and represent unrelated concepts. These results highlight the robustness of AttriCtrl in preserving general generation quality beyond the targeted attribute suppression.

Table 4: Evaluation on unrelated concepts using FID ↓ and CLIP Score ↑.

| Metric | NP | SLD | ESD | Ours |
|---|---|---|---|---|
| CLIP Score ↑ | **0.337** | 0.334 | 0.318 | 0.317 |
| FID ↓ | | 39.322 | 40.016 | 35.665 | **29.963** |

## G  DISCUSSION AND FUTURE WORK

Our experiments are based on the recent FLUX model, a DiT-based architecture. Future work could explore the adaptability of AttriCtrl to U-Net based diffusion models like Stable Diffusion, which would further validate its architectural agnosticism.

As shown in Figure 11, we observe that control becomes less precise when the prompt itself already contains attribute-related modifiers, such as requesting a "hyper-realistic hyperlapse lighting." Quantitatively, the model constrains the attribute intensity within a semantically coherent range, reflecting a prioritization of semantic fidelity over rigid adherence to explicit instructions. This behavior aligns naturally with real-world usage, where user intent is embedded in natural language prompts. Additionally, since our safety dimension is defined relative to the Stable Diffusion safety checker, its effectiveness is inherently bounded by the coverage and biases of this reference model.

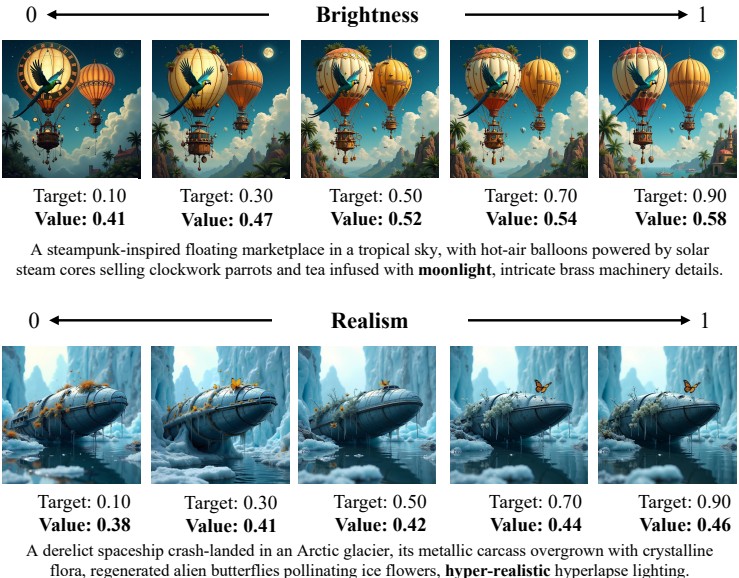

Figure 11: Examples of interaction between prompt semantics and attribute control.

In other words, AttriCtrl does not establish an absolute notion of safety, but instead aligns controllability with a specific, predefined standard.

**Broader Impact and Future Directions.** The contributions of AttriCtrl extend far beyond aesthetic control. The core principle—mapping a normalized scalar value into a dedicated, learnable token sequence via a value encoder—establishes a general and powerful paradigm for fine-grained conditioning in diffusion models. This paves the way for controlling a vast range of previously inaccessible, quantifiable attributes. One can envision future work applying this framework to precisely specify the number of objects in a scene, adjust the geometric properties (e.g., aspect ratio, roundness) of a generated element, or even manipulate abstract physical parameters like simulated temperature or motion blur. Furthermore, our work highlights a promising path toward learning highly disentangled and compositional representations. The ability to independently train and then combine attribute controllers at inference time suggests a future of truly modular, "mixing-console"-like generative systems. This opens up a compelling new research avenue: systematically exploring robust proxy metrics for complex, subjective, or abstract concepts. Devising effective ways to quantify notions like *"creative composition"*, *"emotional tone"*, or *"narrative coherence"* remains a challenging but exciting frontier, for which AttriCtrl provides a foundational control mechanism.

## H MORE EXAMPLES

Figure 12 presents additional qualitative examples of our method controlling the strength of three aesthetic attributes and safety. These results further demonstrate the flexibility and effectiveness of our approach in achieving fine-grained aesthetic control.

## I UNIVERSALITY ACROSS DIFFERENT ARCHITECTURAL MODELS

To demonstrate the universality of AttriCtrl, we present qualitative results generated using three widely adopted diffusion backbones: Stable Diffusion v1.4, Stable Diffusion XL, and Stable Diffusion v3.0. As shown in Figure 13, AttriCtrl consistently enables fine-grained control over target attributes across all three architectures. This indicates that our method can be integrated into various diffusion models with minimal architectural modifications.

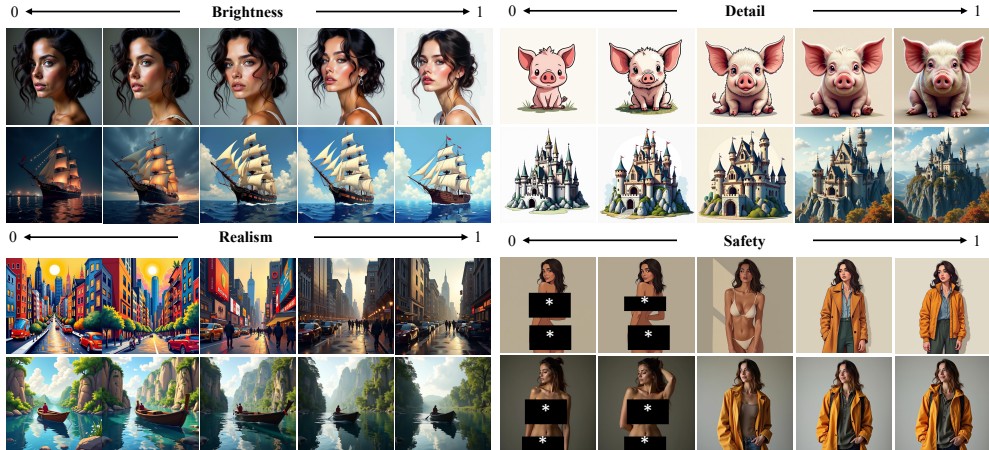

Figure 12: More examples of single attribute intensity control.

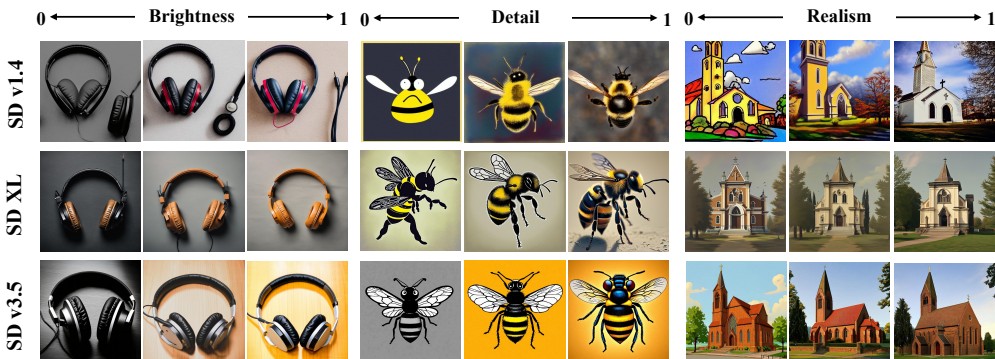

Figure 13: Qualitative results of AttriCtrl across different diffusion architectures.

## J    USE OF LLM

This document was supported by the use of large language models (LLMs), including tools such as ChatGPT and Qwen, during the preparation of this document. These models were used for purposes such as language polishing, improving sentence fluency, and proofreading, only after the core content had been written by the authors. No part of the technical reasoning, data analysis, interpretation of results, or development of ideas was generated or influenced by LLMs. The initial drafts, structure, and key content of all sections were entirely authored by humans. The models were not involved in any decision-making process related to methodology, results, or conclusions.

