# OpenReview forum: "AttriCtrl: A Generalizable Framework for Controlling Semantic Attribute Intensity in Diffusion Models"
_ICLR.cc/2026/Conference — ICLR 2026 Poster_

### Official Review · Reviewer_3gSF · 2025-10-17

**Soundness:** 2
**Presentation:** 2
**Contribution:** 2
**Rating:** 4
**Confidence:** 3

**Summary:**

While diffusion models have been showing really good results in image generation, there might still be cases where they are unable to follow instructions for adjusting visual inputs. This issue arises because text encoders handle words, not continuous values, and current aesthetic alignment methods only capture broad preferences. The proposed framework, AttriCtrl, enables fine-grained, continuous control of aesthetic features by mapping both concrete and abstract traits onto a [0, 1] scale. It uses a plug-and-play value encoder to translate user inputs into interpretable embeddings, allowing smooth and accurate adjustments. AttriCtrl integrates with existing systems like ControlNet while keeping the base model unchanged and computationally efficient.

**Strengths:**

I think the paper has the following strengths:

1) Most of the results seem to be visually nice, which shows that the method can have some good usability.

2) It is very nice that the method can be combined with the likes of ControlNet and Eligen.

3) The paper is quite readable, and a motivated user should be able to understand it.

**Weaknesses:**

I think the paper can be further improved in these aspects.

1) By far my main issue with the paper is the lack of extensive comparisons with the other methods. I understand that for some tasks, this is very hard to do, considering that there aren't clear comparisons. However, for this reason, the authors should have done more evaluations in the task of nudity erasure where there are many other methods and benchmarks. In particular, the authors could have compared with:

[A] Gong et al., Reliable and efficient concept erasure of textto-image diffusion models, 2024

[B] Schramowski et al.,  Safe latent diffusion: Mitigating inappropriate degeneration in diffusion models, 2023

[C] Yoon et al., SAFREE: training-free and adaptive guard for safe text-to-image and video generation, 2024

[D] Lyu et al., One-dimensional adapter to rule them all: Concepts, diffusion models and erasing applications, 2024

[E] Gaintseva et al., CASteer: Steering Diffusion Models for Controllable Generation, 2025

I understand that erasure is not the main focus of the paper, but considering how well-established (and important) it is, it can serve as a proxy on quantitative evaluation of the method.

2) Similarly, for erasure, the authors could have done a user study.

3) Some parts of the paper feel like padding. In particular, almost all equations seem trivial such as doing a normalization, or computing entropy. This does not make the paper be more sophisticated, in fact, it almost have the opposite effect.

4) The authors should have tested their method with more diffusion backbones, currently it is only in Flux. In particular, I would have been very interested to see how it works in SDXL, SD-3/3.5 and SANA.

**Questions:**

I do not have particular questions for the authors, but I would be very interested to see if they can properly address the mentioned weaknesses. If so, I might be willing to increase my score.

---

> ### Author Response · Authors · 2025-11-24
>
> Thank you for the constructive feedback that helped us improve our work. In response to the issues you raised, we have made clarifications below.
>
> **W1**: We sincerely thank you for suggesting these additional baselines. To ensure a rigorous and fair comparison, we invested substantial computational resources to re-train our full AttriCtrl framework on Stable Diffusion v1.4 and evaluated it against all suggested methods—SLD [B], ESD, RECE [A], SAFRE [C], SPM [D], and CASteer[E]—under identical settings.
>
> We report the **removal rate** (RR ↑)  and **generation quality** (with FID↓ for image fidelity and CLIP Score↑ for semantic alignment) in the two tables below. While our method does not achieve the absolute best performance on every metric, it demonstrates consistently **competitive results** across both safety and quality dimensions.
>
> Notably, AttriCtrl enables **fine-grained safety intensity control** at a relatively low computational cost, supporting a spectrum of safety requirements (e.g., from moderate to strict filtering). This represents **a novel direction** in controllable safety mechanisms—one that prioritizes flexibility without requiring retraining. We believe this paradigm opens promising avenues for future work, which can aim to push the upper bound of safety control while maintaining alignment with user intent and visual realism.
>
> | RR in SD v1.4 | SD-NP | SLD[B] | ESD | RECE[A] | SAFREE[C] | SPM[D] | CASteer[E] | Ours |
> | --- | --- | --- | --- | --- | --- | --- | --- | --- |
> | Total(425) | 0.86 | 0.71|0.84| 0.82 |  0.95   |  -0.35   |  0.94   |  0.88   |
> | Buttocks(62) |  0.84     |  0.65     |  0.94     |  0.87     |  0.92     |  -0.58     |  0.95     |  0.87     |
> | Female_Breast(234) |  0.93     |  0.83     |  0.88     |  0.92     |  0.97     |  -0.84     |  0.95     |  0.96     |
> | Female_Genitalia(32) |  0.94     |  0.88     |  0.81     |  0.88     |  0.97     |  -0.19     |  0.94     |  0.88     |
> | Male_Breast(62) |  0.79     |  0.47     |  0.89     |  0.74     |  0.97     |  0.97     |  0.98     |  0.61     |
> | Male_Genitalia(35) |  0.49     |  0.34     |  0.43     |  0.11     |  0.80     |  0.89     |  0.71     |  0.80     |
>
> | Metrics | SD-NP | SLD[B] | ESD | RECE[A] | SAFREE[C] | SPM[D] | CASteer[E] | Ours |
> | --- | --- | --- | --- | --- | --- | --- | --- | --- |
> | FID ↓ |  20.042  |  19.2513  |  19.253  | 19.339 | 20.822 |  19.348  | 26.635 | 20.481 |
> | CLIP Score ↑ |  25.9172   |  25.5302   |  26.3923   |  25.909  |  25.969  |  26.3402   |  25.315  |  25.966  |
>
> **W2**: To evaluate our capability in safety intensity control (not just binary erasure), we conducted **a user study with 10 participants**. Each user was shown images generated at low, medium, and high safety levels and asked to judge whether the safety level increased as intended. The results are as follows. The **97.4% satisfaction rate** demonstrates that users perceive our safety intensity control as effective and meaningful.
>
> | | user1 | user2 | use3 | user4 | user5 | user6 | user7 | user8 | user9 | user10 | Avg |
> | --- | --- | --- | --- | --- | --- | --- | --- | --- | --- | --- | --- |
> | Yes | 98% | 96% | 100% | 100% | 96% | 92% | 98% | 100% | 94% | 100% | 97.4% |
> | No | 2% | 4% | 0% | 0% | 4% | 8% | 2% | 0% | 6% | 0% | 2.6% |

---

> > ### Author Response · Authors · 2025-11-24
> >
> > **W3**:  We agree that clarity and substance should take precedence over notational complexity. In response, we summarize our method’s core pipeline below, integrating the key steps and motivations behind the equations:
> >
> > 1. **Attribute Quantification and Normalization.**
> > For each image ( I ) in the training set, we compute a `raw attribute score`  for the four target properties—Brightness, Detail, Realism, and Safety—using well-established perceptual or model-based metrics (Eqs. 1, 2, 4, and 5).
> > Because these scores originate from heterogeneous distributions, we sample them and normalize them to obtain a `normalized attribute value` $ x^\text{norm}_i \in [0,1] $ via percentile-based rescaling (Eq. 6). This step ensures that user-specified intensity values (e.g., “detail = 0.7”) have consistent semantic meaning across attributes.
> > 2. **Intensity-Aware Conditioning.**
> > During training, the diffusion backbone (e.g., DiT) normally receives two inputs: the text prompt embedding ( $ \mathbf{c} $ ) and the latent image representation ( $ \mathbf{z} $ ).
> > In AttriCtrl, we augment this with a learned intensity representation: the normalized attribute value is passed through a lightweight value encoder to produce an `embedding`  $ \mathbf{v} = \text{ValueEncoder}(x^\text{norm}_i) $ . This $ \mathbf{v} $ is then concatenated with $ \mathbf{c} $ to form an  `enhanced conditioning signal`  $[ \mathbf{c}; \mathbf{v}]$, which guides the denoising process (loss in Eq. 7).
> > 3. **Inference: User-Controlled Generation**
> > At test time, the user specifies only a text prompt and a   `desired intensity`  $ t \in [0,1] $. The value encoder maps  $ \mathbf{t} $ to  $ \mathbf{v} $ , which is injected into the pipeline exactly as during training.
> >
> > Based on the above clarifications, we have further refined the wording and notation in Section 3.
> >
> > **W4**: Following your suggestion to demonstrate **universality across architectures**, we have successfully applied AttriCtrl to SD1.5, SDXL, and the latest SD3.5, in addition to FLUX (see Appendix I for more details).  In all cases, our method achieves smooth and consistent control over attribute intensity, which indicates that our method can be integrated into various diffusion models.

---

### Official Review · Reviewer_LHsn · 2025-10-28

**Soundness:** 2
**Presentation:** 2
**Contribution:** 2
**Rating:** 4
**Confidence:** 4

**Summary:**

This paper addresses the task of controlling specific attributes (e.g., brightness, detail) in image generation. It proposes a method that defines and manipulates these attributes, presenting comparative experiments against other methods like AID. The main contribution lies in formulating this control task and providing a comparative analysis based on the proposed attribute metrics.

**Strengths:**

*   **Interesting Problem:** The core problem identified in Figure 1 is valid and represents an interesting challenge in controllable generation.
*   **Initial Validation:** The experimental results indicate that the proposed method shows some advantages in controlling the defined attributes compared to certain baselines.

**Weaknesses:**

*   **Lack of Conceptual Rigor and Novelty:**
    *   **Inconsistent/Terminological Flaws:** A significant weakness is the lack of conceptual clarity. The title mentions "semantic attributes" while the abstract discusses "aesthetic attributes," which are incorrectly applied to concepts like `detail` and `safety`. The definition of `brightness` (changing object color vs. illumination) is problematic and not aligned with standard understandings in image processing or aesthetics.
    *   **Unnovel Technical Core:** The technical approach for calculating and utilizing the attribute metrics is acknowledged as being based on existing, relatively straightforward techniques (e.g., average pixel intensity for brightness, entropy for detail). The paper fails to clearly articulate what the specific novel technical contribution is beyond the application of these existing metrics.
*   **Inadequate Experimental Validation and Comparisons:**
 Based on Figure 4, I believe the AID method performs the best. AID's results demonstrate precise control over the `brightness`, `detail`, and `realism` attributes, while excellently maintaining content consistency during attribute manipulation.
    *   Regarding `brightness`, AID modifies the ambient lighting rather than the cow's color. This discrepancy is inherently linked to the brightness calculation method used in this paper (average pixel intensity), which does not correspond to illumination in the imaging process.
    *   Using entropy as a complexity metric leads to excessively small objects in low-complexity controls, which is unreasonable from either an aesthetic or user need perspective.
    *   Concerning `realism`, the first result from the proposed method has low realism, whereas all AID results appear highly realistic. The `Diff` metric also does not accurately reflect the corresponding quality.
    *  The experiments in the appendix primarily demonstrate AID's capability in attribute control and semantic consistency. Although cases where AID fails are listed, based on my knowledge of generative models, the proposed method likely also has failure cases. Isolated examples cannot fully substantiate the proposed method's superiority.

    *   **Metrics Lead to Artifacts:** The chosen attribute metrics themselves are a source of limitations, as evidenced by Figure 4, where they cause undesirable effects (e.g., unnaturally small images for low complexity, unrealistic brightness changes). This suggests the core formulation may be flawed.
    *   **Insufficient Evidence for Superiority:** The claim of superiority over AID is not fully convincing. The appendix shows AID failures but does not provide a fair comparative analysis of failure cases for the proposed method. The visual results in Figure 4 suggest AID may perform better in terms of control precision and content preservation.
*   **Issues with Presentation and Reproducibility:**
    *   **Clarity:** The explanation is sometimes unclear, such as the definition and derivation of the final embedding `v`. Variable naming is non-standard (`$normalized_i$`).
    *   **Reproducibility:** The provided code link is broken.
    *   **Scholarly Rigor:** Section 3.2 lacks necessary citations for the techniques used.

**Questions:**

1.  **Conceptual Foundation:** Could the authors clarify the fundamental concept? Are you controlling "semantic" or "aesthetic" attributes? How do your definitions of attributes like `brightness` and `detail` align with or differ from established definitions in image analysis and aesthetic assessment? Specifically, why was average pixel intensity chosen over illumination-aware models for brightness?
2.  **Technical Contribution:** Given that the attribute calculations are based on existing techniques, what is the specific novel technical contribution of the method proposed in this paper? Is it primarily the composition and application pipeline?
3.  **Experimental Comparisons:**
    *   Could you provide a more balanced comparison with AID, including a statistical analysis of performance and failure cases for both methods, rather than just selected examples? The results in Figure 4 seem to favor AID; can you comment on this?
4.  **Metric Design:** The results in Figure 4 show that your metrics can lead to unrealistic outputs (e.g., small image size for low detail, color change for brightness). Do you agree that these metrics have inherent limitations? Have you considered designing or incorporating more sophisticated, perceptually-aligned metrics to address these issues?
5.  **Clarification and Reproducibility:**
    *   How exactly is the final aesthetic intensity embedding `v` obtained? Please provide a clear mathematical formulation or description.

---

> ### Author Response · Authors · 2025-11-24
>
> Thank you for the constructive feedback that helped us improve our work. In response to the issues you raised, we have made clarifications below.
>
> **Q1 / W1-1**:  We appreciate this important clarification request. And we provide the following explanation:
>
> + Most prior work on aesthetics frames aesthetic quality as a single holistic scalar and typically relies on scalar evaluation scores (e.g., ImageReward, PickScore, and HPV) to **estimate that fixed and global score**. In contrast, our work is grounded in the observation that human perceptual preferences **vary across users, contexts, and tasks**.
> + The term “**aesthetic attributes**” in our paper refers specifically to human-perception-aligned visual dimensions that users may wish to adjust continuously based on their needs.
> + We use the phrase “**semantic attributes**” in a broader sense: as any image property that can be quantified by a scalar (e.g., object count, aspect ratio, motion blur, or simulated temperature). As noted in Appendix G, while our primary focus is on aesthetic control, the proposed framework paves the way for controlling a vast range of previously inaccessible, quantifiable semantic attributes.
> + We acknowledge that the current metrics (e.g., HSV Value or Shannon entropy) do not perfectly correspond to the full complexity of human perception—such as physical illumination or intrinsic object detail. Nonetheless, they serve as **effective and tractable proxies for achieving precise numerical control**. Our primary contribution lies in providing a general framework for attribute-intensity control in diffusion models, within which these metrics can be replaced or upgraded when more perceptually aligned estimators become available.  We chose these two metrics for the following reasons:
>     - **Brightness** is operationalized using the Value channel from the HSV color space, which represents perceptual lightness or exposure level—a user-intuitive notion aligned with common photo-editing interfaces. It is a global image-level measure of how “bright” an image appears.
>     - **Detail** is measured using Shannon entropy, which captures perceptual complexity—such as texture richness and edge density. In our training data, minimal entropy values (e.g., 0.004) correspond to extremely sparse and simple visual structures, such as isolated lines or dots. Consequently, when users request such extreme low-detail intensities, the generated outputs faithfully reflect this semantics.
>
> **Q2**: Our work introduces a new perspective on aesthetic and semantic attributes intensity controllability. Towards this goal, we propose a unified framework that goes beyond a simple combination of existing components. The design of our framework naturally decouples attributes, enabling fully independent training and flexible multi-attribute composition at inference time without requiring shared data or joint optimization.
>
> As highlighted in Appendix G, "**our work highlights a promising path toward learning highly disentangled and compositional representations.** The ability to independently train and then combine attribute controllers at inference time suggests a future of truly modular, 'mixing-console'-like generative systems. **This opens up a compelling new research avenue: systematically exploring robust proxy metrics for complex, subjective, or abstract concepts.** Devising effective ways to quantify notions like 'creative composition', 'emotional tone', or 'narrative coherence' remains a challenging but exciting frontier, for which AttriCtrl provides a foundational control mechanism."
>
> **Q3/W2**:  We appreciate your detailed comparison with AID. We acknowledge that in certain qualitative examples in Fig. 4, AID demonstrates strong semantic consistency (e.g., maintaining object scale in low detail, or interpreting brightness as ambient lighting). However, Fig. 4 is specifically designed to evaluate **attribute-match fidelity**—that is, how closely the attribute intensity in the generated image matches the user-specified target value (quantified by the Diff metric). Under this precise criterion, our method achieves significantly smaller Diff values than alternatives.
> In addition, our quantitative results (Table 1) show that **AttriCtrl achieves significantly lower AvgDiff, indicating more precise adherence to the target numerical values across the dataset.** Furthermore, as shown in Fig. 10, AttriCtrl offers a much wider and broader control trajectory compared to AID's limited range. We position AttriCtrl as offering a different trade-off: prioritizing precise, wide-range numerical control over specified metrics, whereas AID may prioritize semantic stability at the cost of control range and precision.

---

> > ### Author Response · Authors · 2025-11-24
> >
> > **Q4**: You are correct that extremely low entropy values can lead to small structures. This is an inherent characteristic of using Shannon entropy as a proxy for 'detail'—minimal entropy corresponds to minimal information, which the model interprets as sparse content. While mathematically correct based on the metric definition, we agree this may not always align with user intent for 'low detail' on a main subject. **This highlights the challenge of metric selection, and our framework allows for integrating more perceptually aligned metrics.**
> > As discussed in Sec. 3.2 (Multi-Attribute Composition), different metrics often produce incompatible statistical distributions, even when they represent the same attribute. If we were to merge datasets and jointly train all value encoders under mixed metrics, **the distributional mismatch would destabilize training and lead to entangled, unreliable control signals**. This is precisely why we adopt a one-metric–one-encoder design, and perform combination only at inference time by concatenating the independently trained control tokens. This design provides stable, disentangled control and is resilient to metric heterogeneity.
> >
> > **Q5**: In Section 3, we detail the procedure for obtaining the final aesthetic-intensity embedding $ v $. For clarity, we summarize the key steps below:
> >
> > 1. **Attribute Quantification and Normalization.**
> > For each image ( I ) in the training set, we compute a `raw attribute score`  for the four target properties—Brightness, Detail, Realism, and Safety—using well-established perceptual or model-based metrics (Eqs. 1, 2, 4, and 5).
> > Because these scores originate from heterogeneous distributions, we sample them and normalize them to obtain a `normalized attribute value` $ x^\text{norm}_i \in [0,1] $ via percentile-based rescaling (Eq. 6). This step ensures that user-specified intensity values (e.g., “detail = 0.7”) have consistent semantic meaning across attributes.
> > 2. **Intensity-Aware Conditioning.**
> > During training, the diffusion backbone (e.g., DiT) normally receives two inputs: the text prompt embedding ( $ \mathbf{c} $ ) and the latent image representation ( $ \mathbf{z} $ ).
> > In AttriCtrl, we augment this with a learned intensity representation: the normalized attribute value is passed through a lightweight value encoder to produce an `embedding`  $ \mathbf{v} = \text{ValueEncoder}(x^\text{norm}_i) $ . This $ \mathbf{v} $ is then concatenated with $ \mathbf{c} $ to form an  `enhanced conditioning signal`  $[ \mathbf{c}; \mathbf{v}]$, which guides the denoising process (loss in Eq. 7).
> > 3. **Inference: User-Controlled Generation**
> > At test time, the user specifies only a text prompt and a   `desired intensity`  $ t \in [0,1] $. The value encoder maps  $ \mathbf{t} $ to  $ \mathbf{v} $ , which is injected into the pipeline exactly as during training.
> >
> > Based on the above clarifications, we have further refined the wording and notation in Section 3.
> >
> > **W3**:  We thank you for pointing out these issues and have addressed them.
> >
> > + We have explicitly described that the normalized attribute value $ x^\text{norm}_i $(called $ \text{normalized}_i $ before) is passed through the value encoder to produce the embedding sequence $ \mathbf{v} = \text{ValueEncoder}(x^\text{norm}_i), $which is then concatenated with the prompt embedding $c$.
> > +  We sincerely apologize for the inconvenience. The anonymous code repository link is currently accessible. We suspect the issue may have been temporary (e.g., due to server caching or transient downtime on the hosting platform).
> > +  We appreciate the reviewer highlighting the need for clearer attribution. Our method is inspired by the T2I-Adapter mechanism, and we have added the appropriate citation in Sec. 3.2.

---

### Official Review · Reviewer_VVmk · 2025-10-31

**Soundness:** 3
**Presentation:** 3
**Contribution:** 3
**Rating:** 6
**Confidence:** 4

**Summary:**

AttriCtrl is a small add-on that lets you directly control how a text-to-image model renders certain qualities—like brightness, detail, realism, or safety—by giving it a numeric value (e.g., brightness = 0.7). It turns that number into a few tokens and appends them to the normal text prompt, while keeping the big diffusion model frozen, so it’s easy to plug in. For concrete attributes, it uses simple measurable definitions (e.g., image brightness); for abstract ones, it relies on CLIP similarity. The method works for one or multiple attributes at once, shows clear, smooth control in examples, and beats prior “prompting” baselines on a main accuracy metric and user preference tests. Overall, it’s a practical, lightweight way to get consistent, continuous control without retraining the main model.

**Strengths:**

- Clear, modular mechanism (frozen base, tiny encoder) that composes with ControlNet/T2I-Adapter; qualitative demos on multi-attribute control are convincing (Figs. 6–7).
- Simple, explicit quantification of attributes (eqs. 1–6) with visualized normalization; practical and reproducible.
- Positive headline numbers: Table 1 shows lower AvgDiff vs AID/Kontext/W-Emb and strong user preference; safety shows higher RR than NP/SLD/ESD (Fig. 5).

**Weaknesses:**

- Robustness under conflicting prompts is acknowledged but not quantified. Appendix G notes degradation when prompts contain strong attribute modifiers (e.g., “hyper-realistic…”) and shows Fig. 11, but lacks a quantified analyses.
- Safety depends on SD checker prior. Safety is defined relative to the Stable Diffusion safety checker; authors note bounded effectiveness (Appendix G). This is a limitation if the checker’s coverage/bias shifts.
- Unrelated-concept quality is only lightly assessed. Table 4 (COCO-10K: CLIP/FID) is encouraging, but the comparison set is safety baselines (NP/SLD/ESD), not control baselines.

**Questions:**

- Could you add an “oracle” numeric-prompt tuning baseline (e.g., prompt templates tuned per attribute) and a CLIP-guided latent interpolation variant to situate AttriCtrl’s gains?
- Since Safety(I) depends on the SD checker prior (eq. 5), can you add an alt-prior (e.g., LAION NSFW classifier) to test sensitivity? Also report RR per category (as in Fig. 5) with confidence intervals.
- Please add control baselines that also aim to steer generation while preserving content regarding Table 4.

---

> ### Author Response · Authors · 2025-11-24
>
> Thank you for the constructive feedback that helped us improve our work. In response to the issues you raised, we have made clarifications below.
>
> **W1**: In response to your request for **quantitative analysis**, we have enhanced **Figure 11 in Appendix G** by annotating both the target attribute values and the corresponding measured values for each generated image. Additionally, we have added* the following **description** in Appendix G: "Quantitatively, the model constrains the attribute intensity within a semantically coherent range, reflecting a prioritization of semantic fidelity over rigid adherence to explicit instructions. "
>
> **Q1**: We have introduced two new baselines as suggested:
>
> + Oracle numeric-prompt tuning baseline: This appends literal textual instructions to the prompt in the form of "{value}% level of [attribute]" (e.g., "70% level of brightness"), directly manipulating the input prompt without additional modules.
> + CLIP-guided latent interpolation variant: For each attribute, we define two extreme prompts (e.g., "0% level of brightness" vs. "100% level of brightness"). At every diffusion step, we compute the corresponding latents and perform weighted interpolation based on the target attribute value.
>
> The average absolute differences (**AvgDiff ↓**) are shown in the following table. In terms of average performance across the three attributes, our method outperforms the two baseline approaches, with **gains** of 0.146 and 0.145, respectively.
>
> | AvgDiff ↓ |  Bright.    | Detail | Realism | Avg.  of Three Attributes |
> | --- | --- | --- | --- | --- |
> | Oracle (numeric prompt) | 0.280 | 0.414 | 0.271 | 0.321 |
> | CLIP-guided latent interpolation| 0.278 | 0.408 | 0.274 | 0.320 |
> | **Ours** | **0.141** | **0.191** | **0.192** | **0.175** |
>
>
> **W2/Q2**: We conduct additional experiments using an alternative safety prior. Specifically, we train a variant of our safety attribute intensity controller using the LAION CLIP-based NSFW detector in place of the default Stable Diffusion Safety Checker.
>
> In the table below, we report the removal rate (**RR ↑**) along with its **95%** **confidence interval** for each exposed body part category. The results indicate that when evaluated with the original **Stable Diffusion Safety Checker** as the metric, our method **achieves higher RR**, suggesting that the Safety Checker provides a more aligned signal for our control objective.
>
> | RR **↑** | **Ours（Safety Checker）** | Ours（LAION NSFW Classifier） |
> | --- | --- | --- |
> | Total(291) | **0.58** | 0.36 |
> | Buttocks(41) | **0.78±0.13** | 0.56±0.15 |
> | Female_Breast(205) | **0.53±0.07** | 0.33±0.06 |
> | Female_Genitalia(2) | **1±0.00** | 0.50±0.69 |
> | Male_Breast(32) | **0.44±0.17** | 0.25±0.15 |
> | Male_Genitalia(11) | **1±0.00** | 0.64±0.28 |
>
>
> **W3/Q3**: To assess the generation quality of the control baselines, we evaluate **FID (↓)** as a proxy for image fidelity and **CLIP Score (↑)** as a measure of semantic alignment, using images generated under varying attribute intensities on COCO-10K. Results are summarized in the table below.
>
> While AttriCtrl does not achieve the highest CLIP Score, it attains the second-best FID among controllable methods that support fine-grained attribute adjustment, demonstrating **a favorable balance between image quality and controllability**. Notably, methods like AID-in/out achieve slightly higher CLIP Scores but at the cost of significantly degraded visual fidelity (as reflected by their higher FID). In contrast, AttriCtrl prioritizes coherent, high-quality image generation while still enabling precise attribute control, which aligns with our core design objective.
>
> | Metrics |  Kontext   |  W-Emb   |  AID-in   |  AID-out   |  Ours   |
> | --- | --- | --- | --- | --- | --- |
> | FID ↓ | 74.813 | 72.212 | 77.381 | 77.392 | 72.293 |
> | CLIP Score ↑ | 27.835 | 26.768 | 28.392 | 28.396 | 27.253 |

---

> > ### Comment · Reviewer_VVmk · 2025-11-25
> > **Comment to the author**
> >
> > Thank you for the clarification and for providing the new results. After carefully considering the overall discussion, I am willing to revise my score accordingly.

---

> > > ### Author Response · Authors · 2025-11-27
> > >
> > > Dear Reviewer VVmk,
> > >
> > > We are pleased to hear that you no longer have concerns. Thank you again for the time and effort you put into reviewing our paper.
> > >
> > > Best regards,
> > >
> > > Authors

---

### Official Review · Reviewer_jx4y · 2025-11-01

**Soundness:** 3
**Presentation:** 3
**Contribution:** 4
**Rating:** 8
**Confidence:** 4

**Summary:**

The paper introduces *AttriCtrl*, a novel method which allows fine-grained control over the strength of a set of textual attributes (specifically *brightness, detail, realism, safety*) in the context of text-to-image diffusion models. Given one attribute a user can decide in a 0-1 scale how strongly that attribute should be present in the final generation. For instance, by selecting the attribute *realism* the user can smoothly interpolate from less realistic to more realistic generations. The method relies on the training of specialized encoders (one for each attribute) which, given a 0-1 score, output a set of tokens which are appended to the textual ones to condition the generation. Experiments are conducted using the FLUX model and prompts from GenEval. Given the novelty of the task of fine-grained attribute control, AttriCtrl is compared to some baselines proposed by this same work and new evaluation metrics are proposed for the task. A user study is conducted to further evaluate the performances. Quantitative and qualitative results as well as the user study generally prove the effectiveness of the proposed method.

**Strengths:**

1. **Original and relevant task.** The paper clearly defines and tackles the underexplored task of *continuous aesthetic intensity control*, introducing a principled framework that goes beyond binary or discrete conditioning.
2. **Sound and well-motivated design.** The use of independent value encoders for each attribute is simple yet effective, allowing plug-and-play control without retraining the diffusion backbone.
3. **Comprehensive evaluation.** The paper presents quantitative, qualitative, and user study evidence to compare over baselines, along with ablations clarifying key design choices. It also provides a clear experimental framework to facilitate future research in this direction.
4. **Clarity and reproducibility.** The paper is clearly written, well structured, and supported by released code and detailed appendices.

**Weaknesses:**

1. **Scalability.** Each attribute requires a separate encoder; while this modularity is elegant, scaling to a larger number of attributes could be computationally expensive as a new encoder must be trained for each attribute.
2. **Content preservation.** In some qualitative examples (e.g., Fig. 4, brightness adjustment of the cow), changes in attribute intensity alter semantic content (the cow becomes white for higher brightness values), which may not be desirable.
3. **Limited backbone diversity.** Experiments are conducted only on FLUX; demonstrating transferability to other architectures (e.g., Stable Diffusion) would strengthen the work.

**Questions:**

1. In line 26, the phrasing *decompose relevant aesthetic attributes* may suggest that AttriCtrl can disentangle and control an arbitrary number of attributes, while in practice it focuses on four (*brightness*, *detail*, *realism*, *safety*). Could this part be clarified or rephrased?
2. How long does the training of each attribute encoder take?
3. Have you experimented also with different models other than FLUX? It would be interesting to understand whether an encoder learnt with FLUX can be used with another model for instance. This would make the method easy to apply by only training the encoders once.
4. Minor notes:
    - The *safety* attribute seems to be missing in the list of line 158.
    - *Ours* should not be capitalized in line 416.

---

> ### Author Response · Authors · 2025-11-24
>
> Thank you for the constructive feedback that helped us improve our work. In response to the issues you raised, we have made clarifications below.
>
> **W1**:  While each attribute requires a dedicated encoder **due to its distinct data distributions**, the **training cost is modest**—each encoder contains only ~18M parameters. This modular design keeps the overhead low.
>
> **W2**: We acknowledge your keen observation regarding content preservation. While our method successfully increases the calculated brightness metric, using simple pixel intensity can sometimes lead to semantic shifts, such as changing the cow's color, rather than just adjusting illumination. This is **a known trade-off** when optimizing for simple, decoupled scalar metrics. However, as detailed in Appendix G, our quantitative analysis shows that **the model generally prioritizes semantic consistency** within a reasonable control range.
>
> **W3/Q3**: Following your suggestion to demonstrate **universality across architectures**, we have successfully applied AttriCtrl to SD1.5, SDXL, and the latest SD3.5, in addition to FLUX (see Appendix I for more details). In all cases, our method achieves smooth and consistent control over attribute intensity, which indicates that our method can be integrated into various diffusion models.
>
> Because different backbones use **different prompt embedding dimensions**, each requires a dedicated value encoder. However, within the same architecture (e.g., FLUX or its fine-tuned variants), the encoder is fully reusable without retraining.
>
> **Q1**:  We have revised line 26 to: "It first **defines** relevant aesthetic attributes, then quantifies them through a hybrid strategy that maps both concrete and abstract dimensions onto a unified [0,1] scale."
>
> **Q2**: Training the FLUX-based version of AttriCtrl takes approximately 20 hours on four NVIDIA A100 GPUs.
>
> **Q4**: Thank you for pointing out these issues. We have made the following revisions:
>
> + Line 158: "We define four semantic attributes that are closely related to human perceptual preferences: brightness, detail, realism, and **safety**."
> + Line 416: "In contrast, **our** AttriCtrl consistently achieves the lowest AvgDiff across all three aesthetic attributes, demonstrating substantially higher control accuracy than all baselines while maintaining smooth content transitions and high image quality."

---

> > ### Comment · Reviewer_jx4y · 2025-11-27
> >
> > Thank you for the clarifications and for running the additional SD experiments. I am happy to maintain my positive score.

---

> > > ### Author Response · Authors · 2025-11-29
> > >
> > > Dear Reviewer jx4y,
> > >
> > > We are pleased to hear that you no longer have concerns. Thank you again for the time and effort you put into reviewing our paper.
> > >
> > > Best regards,
> > >
> > > Authors

---

### Meta-Review · Area_Chair_pXRy · 2026-01-09

**Summary:**

This paper proposes AttriCtrl, a plug-and-play framework for fine-grained, continuous control of attribute intensity (e.g., brightness, detail, realism, safety) in text-to-image diffusion models using lightweight attribute-specific encoders. Reviewer discussion mainly focused on the conceptual definition of attributes, the novelty of the framework beyond proxy metrics, experimental comparisons, and generalization across diffusion backbones.

**Reviewer Concerns:**

Most major concerns were addressed in the rebuttal. The authors clarified the scope and definitions of attributes, strengthened the positioning of the contribution as a general framework for continuous attribute control, added extensive new experiments and baselines (including safety-related methods), conducted user studies, and demonstrated generalization to multiple diffusion backbones. Remaining concerns mainly relate to metric-induced trade-offs, which are acknowledged and do not affect the validity of the method.

**Reviewer Scores:**

Reviewer jx4y: would maintain a high accept score.
Reviewer VVmk: likely to increase to a clear accept.
Reviewer LHsn: likely to increase to borderline accept or weak accept.
Reviewer 3gSF: likely to increase to accept given the added experiments and analyses.

---

### Decision · Program_Chairs · 2026-01-26

Accept (Poster)